# Ultrastable and efficient slight-interlayer-displacement 2D Dion-Jacobson perovskite solar cells

Weichuan Zhang [1,2,3], Ziyuan Liu[4], Lizhi Zhang[4], Hui Wang[4], Chuanxiu Jiang[3,5], Xianxin Wu[3,5], Chuanyun Li[1,6], Shengli Yue[7], Rongsheng Yang[1], Hong Zhang[1], Jianqi Zhang [1], Xinfeng Liu [3,5], Yuan Zhang[7] & Huiqiong Zhou [1,3] ✉

Stability has been a long-standing concern for solution-processed perovskite photovoltaics and their practical applications. However, stable perovskite materials for photovoltaic remain insufficient to date. Here we demonstrate a series of ultrastable Dion−Jacobson (DJ) perovskites (1,4-cyclohexanedimethanammonium)(methylammonium)$_{n-1}$Pb$_n$I$_{3n+1}$ ($n \geq 1$) for photovoltaic applications. The scalable technology by blade-coated solar cells for the designed DJ perovskites (nominal $n = 5$) achieves a maximum stabilized power conversion efficiency (PCE) of 19.11% under an environmental atmosphere. Un-encapsulated cells by blade-coated technology retain 92% of their initial efficiencies for over 4000 hours under ~90% relative humidity (RH) aging conditions. More importantly, these cells also exhibit remarkable thermal (85 °C) and operational stability, which shows negligible efficiency loss after exceeding 5000-hour heat treatment or after operation at maximum power point (MPP) exceeding 6000 hours at 45 °C under a 100 mW cm$^{-2}$ continuous light illumination.

Service life has been a long-standing concern for perovskite photovoltaics[1–6]. The lifetime of perovskite materials and the corresponding devices is reported to be affected by numerous factors, which can be summarized into two main categories: intrinsic (hygroscopicity, strain, and ion migration, etc.) and extrinsic factors (moisture, oxygen, light irradiation, heat, and electric fields, etc.)[7,8]. To overcome these looming stability issues, strategies including encapsulation techniques, additive engineering, interfacial engineering, composition engineering, and materials design and dimensional regulation are constantly being developed and applied[4,9–11]. Encapsulation techniques can prevent the sensitivity factors of humidity and oxygen from affecting the material stability of perovskites[9]. Additive and composition engineering can be applied to tune the strain, decomposition barrier, or energy, thereby improving the films and their device stability[12–15]. Instability caused by the interface defects and ion migration can be further improved via the interfacial engineering strategy[16–18]. It is found that these constructive strategies tend to focus on preventing the unstable factors from affecting the materials, while the inherent instability of the perovskites can be further optimized via material design and dimensional regulation[19,20]. Although great efforts have been made, challenges remain in improving the long-term stability of perovskites and their devices.

In this work, we present a series of ultrastable DJ perovskites (CDMA)(MA)$_{n-1}$Pb$_n$I$_{3n+1}$ (CDMA = 1,4-cyclohexanedimethanammonium,

[1]CAS Key Laboratory of Nano system and Hierarchical Fabrication, National Center for Nanoscience and Technology, 100190 Beijing, PR China. [2]School of Electrical Engineering, University of South China, Hengyang 421001 Hunan, PR China. [3]University of Chinese Academy of Sciences, 100049 Beijing, PR China. [4]Laboratory of Theoretical and Computational Nanoscience, National Center for Nanoscience and Technology, Chinese Academy of Sciences, 100190 Beijing, PR China. [5]CAS Key Laboratory of Standardization and Measurement for Nanotechnology, National Center for Nanoscience and Technology, 100190 Beijing, PR China. [6]College of Chemistry and Materials Engineering, Beijing Technology And Business University, 100048 Beijing, PR China. [7]Beijing Advanced Innovation Center for Biomedical Engineering, Beihang University, 100191 Beijing, PR China. ✉e-mail: zhouhq@nanoctr.cn

$n \geq 1$) for stable and efficient solar cells. Devices based on the nominal $n = 5$ composition achieved a PCE of 19.11% via the scalable blade-coating method under the atmospheric environment, comparable to that of DJ perovskite under the spin-coating technique. Importantly, these cells exhibit dramatic moisture (90% RH), thermal (85 °C), and operational (MPP tracking) stability, where the corresponding degradation ratios are only 8% for moisture and negligible efficiency loss for thermal and operational stability, respectively, within the aging time exceeds 4000, 5000 and 6000 h. Structural analyses show that these materials have an extraordinary slight interlayer-displacement quantum-well configuration that differs from most DJ perovskites, which can reduce interlayer spaces and tune their mutual alignment to facilitate interlayer charge transport and structural stability. In addition, the use of cycloalkyl organic cations as interlayer cations can maintain the flexibility and electronegativity of the molecule compared with alkyl-chain and aryl organic cations, which is beneficial to reducing lattice stress and enhancing structural stability.

## Results

### Performances of the DJ perovskite solar cells

Two-dimensional (2D) perovskites have received increasing attention owing to their remarkable inherent structural stability and desirable photophysical properties compared to their three-dimensional (3D) counterparts[4,13,21–25]. In contrast to Ruddlesden–Popper (RP) perovskites, DJ types with ditopic diammonium cations are deemed to strengthen the connection between inorganic layers and increase the overall structural rigidity, thereby possibly enhancing stability[26–28]. Especially in recent years, substantial efforts have also been devoted to developing the performances of DJ perovskite solar cells. However, most of these promising DJ perovskite did not show the expected improvement in stability[27–34]. Studies on RP and DJ series even show that DJ perovskites are relatively less stable in atmospheric environments[30]. For example, the 1,4-phenylenedimethanammonium- and *m*-phenylenediammonium-based DJ perovskites exhibit poor moisture stability within a few hours (Supplementary Figs. 1 and 2). Therefore, systematic research on the key factors of stability in DJ materials and guidance for the design of stable DJ perovskites are highly desirable for fabricating stable and efficient scalable solar cells.

To demonstrate the prospects of the designed DJ perovskite series toward scale solar cells, we fabricated the solar cells (Supplementary Figs. 3 and 4 and Supplementary Table 1) under an atmospheric environment using a scalable deposition technique of blade coating, as illustrated in Fig. 1a. The reference solar cells adopt the nominal $n = 5$ PDMA DJ perovskite with a similar organic molecular configuration to CDMA materials, where the PDMA has been widely studied for improving device stability recently. Figure 1b depicts the current density–voltage ($J$-$V$) curves of the DJ perovskite solar cells. The reference devices have a maximum PCE of 14.87% with an open-circuit voltage ($V_{OC}$) of 1.06 V, a short-circuit current density ($J_{SC}$) of 18.32 mA cm$^{-2}$, and a fill factor of 76.46%. The nominal $n = 5$ CDMA-based perovskite solar cells show a maximum PCE of 19.11% with a $V_{OC}$ of 1.16 V, a $J_{SC}$ of 20.41 mA cm$^{-2}$, and a fill factor of 80.56%. To our best knowledge, the PCE of CDMA perovskites is the highest efficiency for the blade-coating 2D perovskite solar cells reported to date (Fig. 1b inset and Supplementary Table 2). It is comparable to most of the reported 2D DJ perovskite solar cells by spin-coating technology. External quantum efficiency (EQE) was further performed to confirm the measured $J_{SC}$ (Supplementary Fig. 5). It shows that the EQE value of the PDMA device is lower than the CDMA solar cell over the whole visible-light absorption region with integrated $J_{SC}$ of 17.57 and 19.58 mA cm$^{-2}$, respectively, which reasonably match the measured $J_{SC}$ under the solar simulator.

Having demonstrated excellent device performances, we have further studied the stability of complete solar cells under moisture, heat, and light stressing. We first determined their prominent moisture

stability by storing the unencapsulated solar cells in a constant-temperature humidity chamber with ~90% RH at ~22 °C. Figure 1c depicts that the PCE of the PDMA-based solar cells rapidly decreases to about zero after roughly another 500 h of aging, while the CDMA-based devices remain around 92% of their initial efficiency after 4394 h of aging. By comparing the aged devices, we find that the active layer of the PDMA-based devices has become colorless, while the CDMA-based device shows no visible discolouration (Fig. 1c inset). It is confirmed that the outstanding humidity tolerance of the CDMA-based DJ perovskites. We also investigated the thermal stability of unencapsulated perovskite solar cells on a hotplate at 85 °C (Fig. 1d). After 3000 hours of aging, the PDMA-based perovskite solar cells show a 50% decline in their initial efficiency, while the CDMA-based devices show a negligible efficiency loss exceeding 5000 h. It also verifies that the CDMA perovskites have a superior thermal tolerance. We further investigated the operational stability of the unencapsulated DJ solar cells under a nitrogen-atmosphere glovebox, using MPP tracking under a simulated 1-sun white light-emitting diode (LED) illumination at ~45 °C. Figure 1e shows the long-term operational stability of the perovskite solar cells. The PCE of the PDMA-based devices was reduced by about 30%, whereas the CDMA-based solar cells only show a slight fluctuation around their initial efficiency over 6000 h aging without a downward trend. As depicted in Fig. 1f–h, the stability of the CDMA-based DJ series, to the best of our knowledge, is almost the best among the 2D and 3D perovskite solar cells (summarized in Supplementary Table 3).

### Device characteristics

Given the remarkable device performances of the designed DJ series, we further examined the effect of device characteristics on photovoltaic performance and stability. The stabilized power output measured at a MPP for the DJ perovskite solar cells was performed, as shown in Fig. 2a. The $J_{SC}$ of the PDMA perovskite solar cells exhibits a slight increase, while the CDMA is almost unchanged within 600 s, implying a potential enhancement in operational stability. The best-performing perovskite solar cells devices were also measured at the MPP to obtain a stabilized PCE of about 19% (Fig. 2b). Furthermore, Fig. 2c shows a statistical distribution of the calculated hysteresis factor from the measured PCE of the PDMA and CDMA perovskite solar cells. It reveals that the CDMA perovskite solar cells have a relatively lower and stable-distributed hysteresis, which could be conducive to fabricating stable and scalable solar cells toward practical application.

Figure 2d and Supplementary Fig. 6 show the photovoltage and photocurrent decay kinetics of the CDMA- and PDMA-based solar cells, respectively. It shows that the CDMA-based films exhibit a longer photovoltage/shorter photocurrent decay times of 129.2 ns/26.3 μs compared to the PDMA-based perovskite solar cell (105.1 ns/18.5 μs). The relatively long photovoltage and fast photocurrent decays in the CDMA-based device suggest that it effectively reduces recombination and improves charge extraction[35]. Besides, the recombination kinetics were further investigated by the light-intensity dependent $V_{OC}$ and $J_{SC}$. Supplementary Fig. 7 depicts a linear relationship between light intensity ($I$) and $J_{SC}$ ($J \propto I^\alpha$) for the PDMA- and CDMA-based perovskite solar cells. The CDMA-based device has a favorably fitted α value of approximately 1, indicating negligible bimolecular recombination and good charge transport. Figure 2e depicts a linear relationship between $V_{OC}$ and the logarithm of light intensity. It is fitted by the equation $V_{OC} \propto nkT/q \cdot \ln(I)$, where the $n$ is the ideality factor, $k$ is the Boltzmann constant, and $q$ is the electron charge. The CDMA cell shows a slope of 1.70 $kT/q$, higher than the PDMA device of 1.93 $kT/q$, implying the inhibited trap-assisted recombination, which might be beneficial to improving the fill factor[1]. These notable device performances suggest that the CDMA-based DJ perovskites have great potential for fabricating stable solar cells.

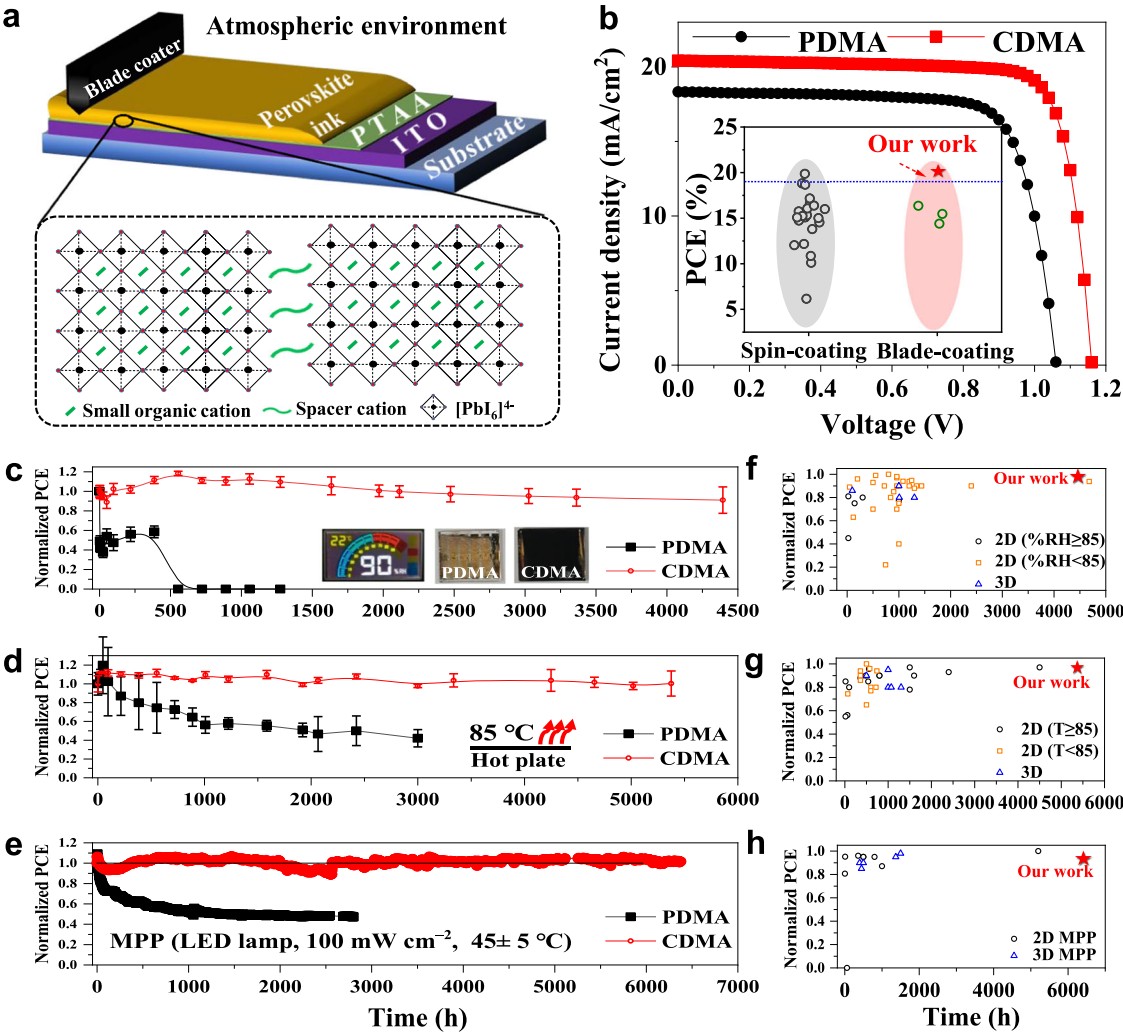

**Fig. 1 | Device performance and stability. a** Schematic illustration of the blade-coating film and the corresponding device configuration under atmospheric environment at room temperature. A schematic diagram of the n = 5 DJ perovskite structure is shown in the dashed box. **b** J-V curves of the best-performance PDMA and CDMA-based solar cells with p-i-n structure. Inset: Comparison of the photovoltaic performance of recently published 2D perovskite solar cells. The gray and light red shadows represent the spin- and blade-coating devices, respectively. **c** Normalized PCEs of the unencapsulated solar cells stored in a constant temperature humidity chamber (~90% RH). The error bars represent the standard errors. Inset: Image of the experimental instrument setup parameter and comparison of device back side after 4000 h. **d** Thermal stability of solar cells treating at an 85 °C hotplate in N$_2$ atmosphere. The error bars represent the standard errors. **e** Maximum power point tracking measured for the perovskite solar cells at ~45 °C in N$_2$ atmosphere. Summary of the stability characterization (**f**: moisture stability; **g**: thermal stability; **h**: MPP &light stability) of most of the reported 2D perovskite solar cells, compared with recent reported typical highly efficient and stable 3D cells. (see the details in Supplementary Table 3 in the Supporting Information).

Steady-state photoluminescence (PL) and time-resolved PL measurements were further performed to explore the optoelectronic properties of the perovskite films. Supplementary Fig. 8 shows the PL spectra and decay of DJ perovskites on the PTAA transport layer. The fit data are summarized in Supplementary Table 4. Remarkable PL quenching is observed on the CDMA-based perovskite film, indicating efficient charge transfer from the active layer to the transport layer. In addition, it is shown that the decay times of CDMA-based perovskite films are smaller than those of PDMA, which suggests that the CDMA active layer has a relatively high charge transfer ability from perovskite to PTAA layer[36,37]. Moreover, higher charge mobilities and lower trap densities were further observed in the CDMA-based perovskites based on space charge limited current (SCLC) measurements in the hole- and electron-only devices (Fig. 2f, and Supplementary Figs. 9 and 10). The low trap density indicates that the CDMA-based perovskite films have a relatively low defect density and suppress the trap-assisted non-radiative recombination losses. Besides, the lower dark current curve of CDMA-based devices further confirmed the low leakage current

density and carrier recombination, which contributes to fabricating high-performance devices (Supplementary Fig. 11). In addition, we also evaluated the potential of the DJ series in optoelectronic and photovoltaic. Density functional theory (DFT) calculations and energy level diagrams of the films show that n ≥ 3 perovskites have well-matched calculated band gaps and energy level alignments for photovoltaic applications (Supplementary Figs. 12–16). These features suggest that the synthesized DJ series have enormous potential as stable light absorbers for applications in solar cells.

## Characterization of the perovskite films

Top-view scanning electron microscopy (SEM) was further studied for the morphology of the films. As shown in Fig. 3a, the PDMA-based perovskite film shows a uniform grain distribution with small sizes. Differently, CDMA-based film exhibits a distinct enlargement of grain sizes, and some small 2D wafers cover the corresponding grain boundaries (Fig. 3b), which probably involve self-passivation of the interface for the stability[38]. XRD measurements and ultraviolet-visible

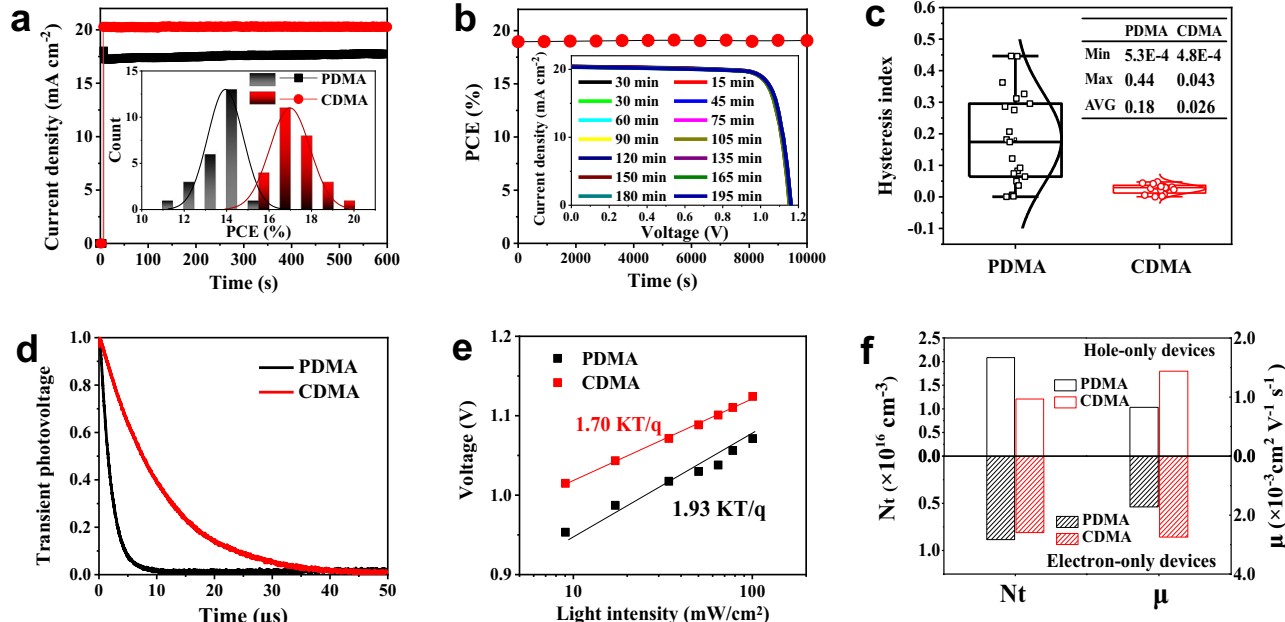

**Fig. 2 | Characterization of the device performances. a** Steady-state current density, measured for 600 s at the MPP. Inset: Histograms showing the device PCEs, fitted with Gaussian distributions (solid lines). **b** Stabilized power output at the MPP for the best-performing perovskite solar cells. Inset: *J–V* curves of the device with different delay times. **c** Statistical comparison of hysteresis factors (20 devices, respectively). The median value, maximum and minimum values, and 25% to 75% region of data are represented by the horizontal line (across the box), top/bottom bars, and box, respectively. Circle symbols represent data distribution. Inset: Table of the maximum, minimum, and average value of hysteresis factors. **d** Photovoltage decay curves of the corresponding devices. **e** Light intensity dependent voltage values of the devices on a seminatural logarithmic scale. **f** Carrier mobility and trap density comparison of electron-only and hole-only devices based on the PDMA- and CDMA-based 2D perovskite films.

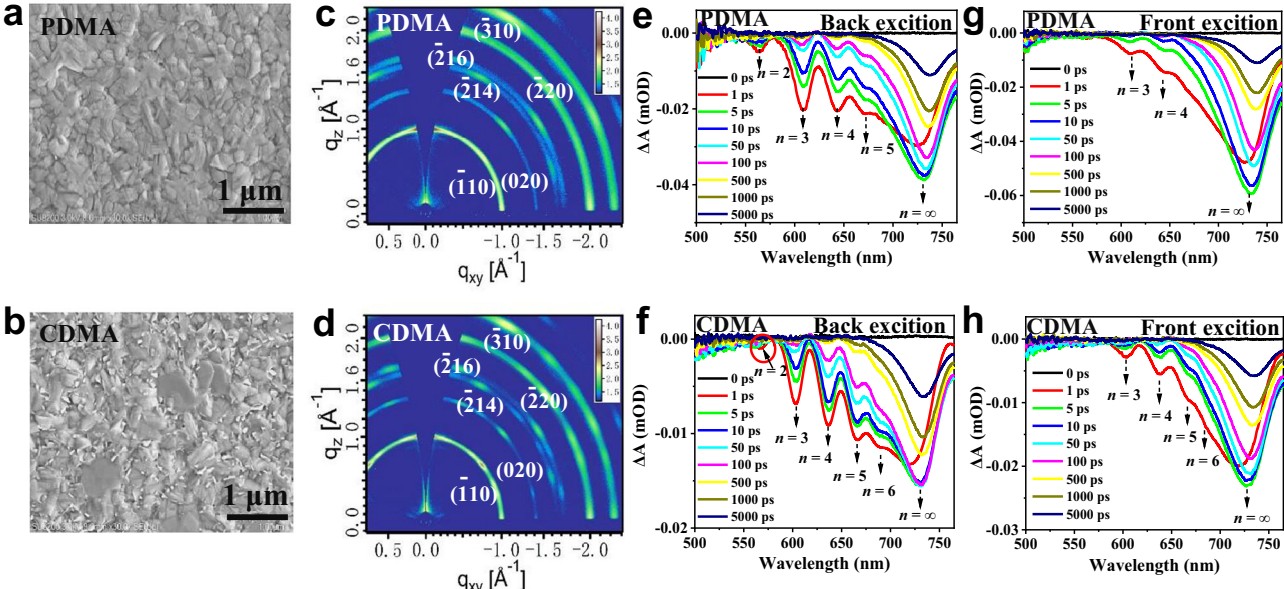

**Fig. 3 | Characterization of the nominal *n* = 5 perovskite films. a, b** SEM images of the PDMA- and CDMA-based perovskite films, respectively. **c, d** GIWAXS images of the DJ perovskite films. **e, g** TA spectra under different delay times for the PDMA film under back- and front-photoexcitation. **f, h** TA spectra under different delay times for the CDMA film under back- and front-photoexcitation.

(UV–Vis) absorption spectra of blade-coating films were depicted in Supplementary Fig. 17a. Compared to the PDMA-based films, CDMA-based films exhibit a relatively narrower full width at half maximum (FWHM) and stronger intensity of diffraction peaks, suggesting improved crystallinity and enlarged grain sizes according to Scherrer's equation (Supplementary Fig. 17b)[39]. The UV–Vis absorption spectra demonstrate that both DJ perovskite films have evident 2D phase distributions (Supplementary Fig. 17c). In contrast to the PDMA-based film, the 2D absorbing region of the CDMA sample has a relatively high intensity, suggesting the relatively abundant 2D phase distribution is beneficial to its shortwave absorption and film stability. The grazing incidence wide-angle X-ray scattering (GIWAXS) were further studied, as shown in Fig. 3c, d. Compared with PDMA-based perovskite, the CDMA-based film depicts relatively strong and discrete Bragg

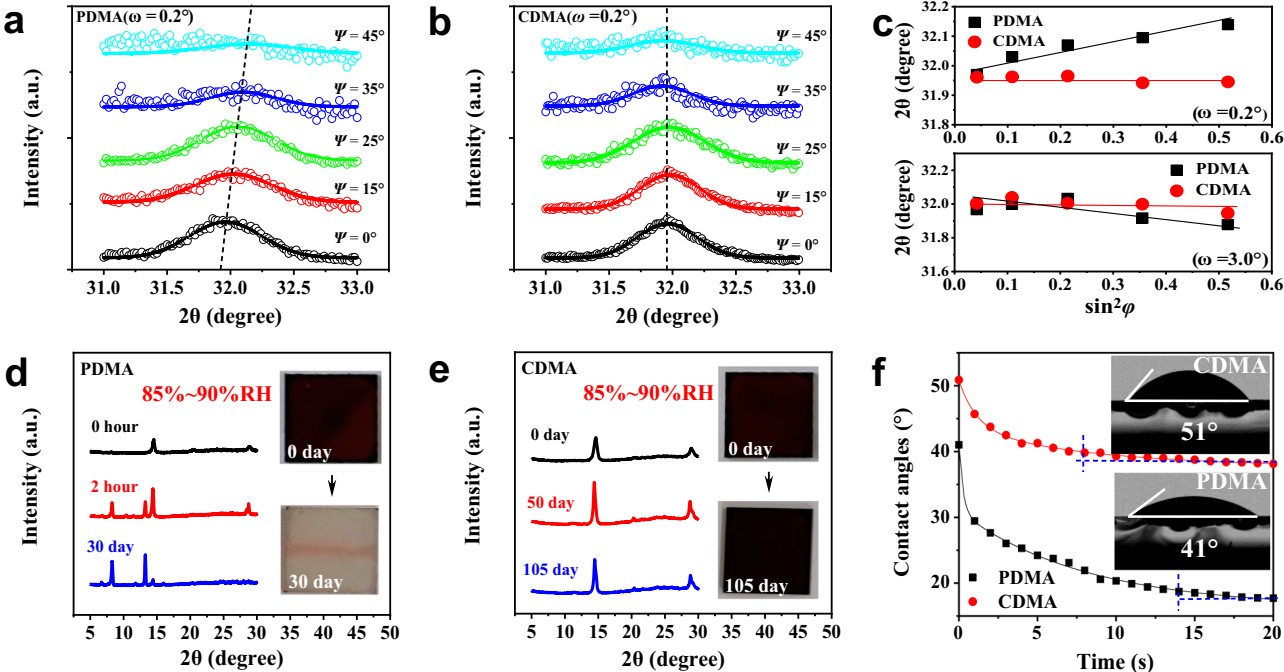

**Fig. 4 | Strain distribution and stability measurements. a, b** GIXRD spectra at different tilt angles for the PDMA- and CDMA-based film, respectively. **c** Linear fit of $2\theta$-$\sin^2\varphi$ in the different regions for the PDMA- and CDMA-based film. **d, e** XRD patterns of films exposed in a constant temperature humidity chamber with relative humidity (RH) of 85–90% under dark at room temperature. Inset: Comparison of the fresh perovskite films and after aging. **f** Dynamic dissolving process of a water droplet on the surface of PDMA- and CDMA-based DJ perovskite films. Inset: Contact angles of the water droplets on perovskite films.

spots, which indicates the CDMA film has better crystallization, benefiting charge transport. The specific orientations assigned as (110), (020), (214), (216), (220), and (310) for both PDMA and CDMA-based DJ perovskites. Azimuthal scans of the (110) and (220) diffraction were further extracted from the GIWAXS measurements (Supplementary Fig. 18).

Femtosecond transient absorption (TA) measurements were further conducted to investigate the phase distribution and energy transfer process on the stability and performance of devices (Supplementary Fig. 19). Figure 3e, f represent the TA spectra of the PDMA- and CDMA-based perovskites excited from the backside. Both films feature distinct ground-state bleach peaks at 608, 643, 671, and 730 nm, corresponding to the $n = 3$, 4, 5, and 3D-like phases, respectively, in accordance with the steady-state absorption. Differently, there is a distinct 2D perovskite phase with $n = 2$ in PDMA, and the intensity of the bleaching peaks shows that $n = 3$ is the strongest in the 2D perovskite phases. By comparison, the CDMA films have an additional high $n$-value 2D phase ($n = 6$), and their bleaching peak intensities are enhanced with the increased perovskite layers. Particularly, the peak of $n = 2$ in CDMA-based perovskite films is extraordinarily inconspicuous and almost undetected. Due to the low-$n$ perovskites ($n < 3$) mismatch in the energy level alignments with the charge transport layers (see Supplementary Fig. of energy level alignments within Supplementary Information), the phase distributions of the CDMA-based film suggest that the distributions are more reasonable and beneficial to charge-carrier transfer and device stability. Under frontal excitation, PDMA film exhibits a dominating bleaching peak at 730 nm, corresponding to the 3D-like phase, accompanied by several unsharp peaks of the low-$n$ DJ perovskite phase (Fig. 3g). However, CDMA-based films can observe a more obvious 2D bleaching peak, indicating that the distribution of 2D phases is more reasonable in the active layer, which is helpful to the stability of the device (Fig. 3h). In addition, the PDMA film exhibits a decay times of $\tau = 0.24$, 0.53, 1.86, and 4.52 ps ($n = 2$-5), respectively, whereas the CDMA film shows relatively faster decay times of $\tau = 0.21$, 0.42, 0.53, and 1.86 ps for the

phases with $n = 3$, 4, 5, and 6, respectively (Supplementary Fig. 20 and Supplementary Table 5). It indicates more efficient energy transfer from 2D- to 3D-like phases in CDMA films[6].

## Characterization of strain and stability

In contrast to the perfect lattice arrangement in a single crystal, residual strains are commonly observed in thin films of perovskites, including interfacial-residual stress, lattice mismatch, etc[40,41]. We picked up two depths to investigate the residual strain in the PDMA- and CDMA-based perovskite films according to the phase distribution shown by the TA characterization. It includes a shallow region near the surface dominated by 3D-like phases and a relatively deeper district with a relatively abundant distribution of 2D perovskites. Figure 4a depicts the grazing incident X-ray diffraction (GIXRD) of PDMA-based perovskite film relative to the (310) crystal plane[40]. The diffraction peaks gradually shifted to the right side with the increase of $\Psi$ (0°–45°), revealing a decrease in crystal plane distance and that the film is bearing compressive strain. For the region rich in 2D perovskites, PDMA films exhibit a trend shift to the left with the increase in $\Psi$ values (Supplementary Fig. 21a). However, the trend of variation slightly deviates from the linear direction of fitting, indicating an inhomogeneous distribution of 2D perovskite phases, which is detrimental to the device performance and stability. Differently, the CDMA-based films exhibit negligible displacement with the increase of $\Psi$ in both regions (Fig. 4b and Supplementary Fig. 21b). These results are also consistent with the residual strain calculated via Williamson–Hall plots (Supplementary Fig. 21c). It suggests that the fabricated DJ perovskite film is almost free of residual strain, contributing to device stability. The linear fitting of $2\theta$ and $\sin^2\varphi$ at both regions is depicted in Fig. 4c.

Then, we further examine the stability of the materials and the corresponding films. XRD measurements of crystal powder samples stored over 300 days in the atmospheric environment (Supplementary Fig. 22) indicate the remarkable stability of designed DJ perovskites. Moisture stability of the films was further tracked under an 85-90% RH

constant humidity chamber at room temperature. It is found that the nominal $n = 5$ PDMA-based perovskite films exhibit poor moisture stability within a few hours (Fig. 4d). After a month of storage, the black phase of the film has been completely converted and decomposed into a broad bandgap phase (Fig. 4d inset). In contrast, the CDMA-based DJ perovskite films exhibit remarkable moisture stability without the appearance of impurity diffraction peaks after storage exceeding 100 days in the same condition (Fig. 4e). In addition, more characterizations of film stability, such as thermal and light stability, were further performed, as shown in Supplementary Figs. 23–28, suggesting the great potential of the CDMA-based DJ series for photovoltaics. We further examine the water-resistant performance of films by measuring water contact angles on the film surface (Supplementary Fig. 29). As shown in the inset of Fig. 4f, the CDMA-based film exhibits a larger water contact angle (51°) than the PDMA-based film (41°), indicating a relatively high water-resistant performance. In particular, the dynamic dissolving process of a water droplet on the film surface of the CDMA-based perovskite film and its color did not change significantly within 10 s (Supplementary Movie 1). It is worth mentioning that the water droplet experiment was performed under a recycled CDMA film, which appears to be unaffected by the last droplet experiment. However, the fresh PDMA film shows a distinct decomposition behavior within 3 s, where the droplet rapidly spreads out, and the film turns white in color (Supplementary Movie 2). The rapid decomposition behavior of PDMA films could be ascribed to the formation of 1D hydrate (PDMA) $Pb_2I_6 \cdot 2H_2O$ with water molecules[31]. The dynamic contact-angle curves are shown in Fig. 4f, which confirms that CDMA-based perovskite films have excellent waterlogging resistance. These features suggest that the CDMA DJ series have enormous potential as stable light absorbers for photovoltaic applications.

## Structural analyses

Structurally, multi-quantum-well 2D perovskites possess two typical geometry configurations, with the inorganic layers enabling them to stack in an aligned or a half-displacement arrangement[4,20]. The RP perovskites are dominated by the unaligned configuration, possessing pairs of interdigitated interlayer spacers[20]. In contrast to RP perovskites, DJ types with ditopic diammonium cations are deemed to strengthen the connection between inorganic layers and increase the overall structural rigidity, thereby possibly enhancing stability[26–28]. Moreover, the short distance between the layers can contribute to charge transport[29]. On this basis, we further synthesized and examined the structures of the designed DJ perovskites by single-crystal XRD analyses (Supplementary Data 1, 2, Supplementary Figs. 30–35, and Supplementary Tables 6–20). Structural analyses demonstrate that the application of CDMA cations can construct a series of 2D quantum-well analogs (Fig. 5a), as previously reported in other typical DJ perovskites[4]. Differently, the inorganic frameworks of these CDMA-based perovskite structures exhibit slightly offset from traditional DJ perovskite materials (0, 0; 0, 1/2; or 1/2, 1/2 displacement). The displacement distances are 1.097, 1.005, and 0.955 Å, respectively (Fig. 5b), which are slightly smaller with the increase in the layer number of 2D perovskite layers. In contrast, the inorganic segments of the PDMA-based crystal structure (CCDC numbers 2078953 and 2078954) assemble in the typical 0-displacement configuration, as shown in Fig. 5c, d.

We further measured the size of these two cations, in which the CDMA and PDMA cations show lengths of 7.7 and 7.4 Å, respectively (Fig. 5e). In general, 2D perovskites with longer-dimensional cations have larger layer spacing. However, the slight-interlayer-displacement CDMA perovskites with longer cation lengths show shorter layer spacing, slightly decreasing with the increased layer number of 2D perovskite layers (5.8, 5.5, and 5.3 Å for $n = 1–3$). The I···I distances between the adjacent inorganic layers for $n = 1–3$ perovskites are 5.89, 5.63, and 5.48 Å, respectively (Supplementary Fig. 36). Differently, layer spaces

of aligned PDMA perovskites are 6.0 Å ($n = 1$) and 5.8 Å ($n = 2$). The schematic diagram to illustrate the difference between the slight-interlayer-displacement and 0-displacement DJ perovskite series is shown in Fig. 5f. Distinctly, decreasing interlayer spaces and tuning their mutual alignment are significant parameters for controlling the optoelectronic and electrostatic properties as they strengthen electronic interactions[29] and strain[42] between the inorganic layers, and consequently, facilitate interlayer charge transport and structural stability.

In addition to the studies on the slightly decreased layer distance, we further investigated the microscopic mechanism facilitating the favorable stability by the large cations. For the 2D perovskites, large spacer cations connect adjacent inorganic perovskite slabs via hydrogen bonding, which has been recognized as the key to improving structural stability[43]. Therefore, we extracted the data on hydrogen bonds between space cations and inorganic slabs of $n = 1$ and $n = 2$ 2D perovskites based on the obtained CDMA and PDMA crystal structures, as shown in Supplementary Fig. 37. Statistical comparisons of their corresponding distributions were depicted in Fig. 5g, h. It shows that the average hydrogen bond lengths of both $n = 1$ and $n = 2$ PDMA-based perovskites are larger than those of CDMA-based materials, suggesting a relatively poor interaction in relation to structural stability. It is consistent with the displacement of the infrared results (Supplementary Fig. 38). Moreover, the equatorial Pb-I-Pb angles that interacted with the PDMA cations are smaller than those of CDMA-based perovskites (Supplementary Fig. 39), indicating notable twist features that could seriously affect structural stability. The weak hydrogen-bond interaction and relatively twisted equatorial Pb-I-Pb angles could be the reason that the PDMA-based perovskites easily form a 1D hydrate (PDMA) $Pb_2I_6 \cdot 2H_2O$ by reacting with water molecules[31]. In contrast, the relatively strong hydrogen-bond interaction of CDMA cations can strengthen the connection between inorganic layers, thereby resisting the attack of water molecules, protecting the inorganic perovskite layers, and significantly improving stability. In addition to structural analyses, the formation energies of the reaction between perovskite and water molecules were further evaluated via the DFT calculations (Supplementary Data 3 and Supplementary Fig. 40). For the PDMA-based perovskite, $\Delta E_{PDMA} = E_{Product} - E_{Reactant} = -0.4$ eV, where the total energy of the reactant is higher than the total energy of the product, indicating that $(PDMA)PbI_4$ can spontaneously react with water to synthesize $(PDMA)Pb_2I_6 \cdot 2H_2O$ and $(PDMA)I_2$ with lower total energy. By contrast, the CDMA-based results show the total energy of the reactant is lower than the total energy of the product ($\Delta E_{CDMA} = E_{Product} - E_{Reactant} = 0.472$ eV), indicating that $(CDMA)PbI_4$ cannot react spontaneously with water and is difficult to synthesize $(CDMA)Pb_2I_6 \cdot 2H_2O$ experimentally. These calculations further demonstrate that CDMA-based perovskites have excellent stability.

## Discussion

In summary, we have found that the interlayer-displacement DJ perovskite configuration with flexible organic cations can contribute to structural stability. On this basis, we precisely designed a series of interlayer-displacement DJ perovskites with excellent material stability, as expected. Moving forward, we applied these potential materials to the scalable blade-coating process and achieved an efficiency exceeding 19.11%. Importantly, fabricated cells also exhibit remarkable humidity, thermal, and operational stability. After keeping them in a 90% RH or 85 °C continuous aging condition for over 4000 h and 5000 h, respectively, the devices show an 8% degradation for the humidity stability test and negligible efficiency loss for thermal stability measurement. Particularly, the operational stability under continuous light stress (MPP tracking) shows negligible efficiency loss exceeding 6000 h. The designed interlayer-displacement DJ series provides a significant and potential pathway for constructing new structurally stable 2D perovskites. The blade-coating application of the

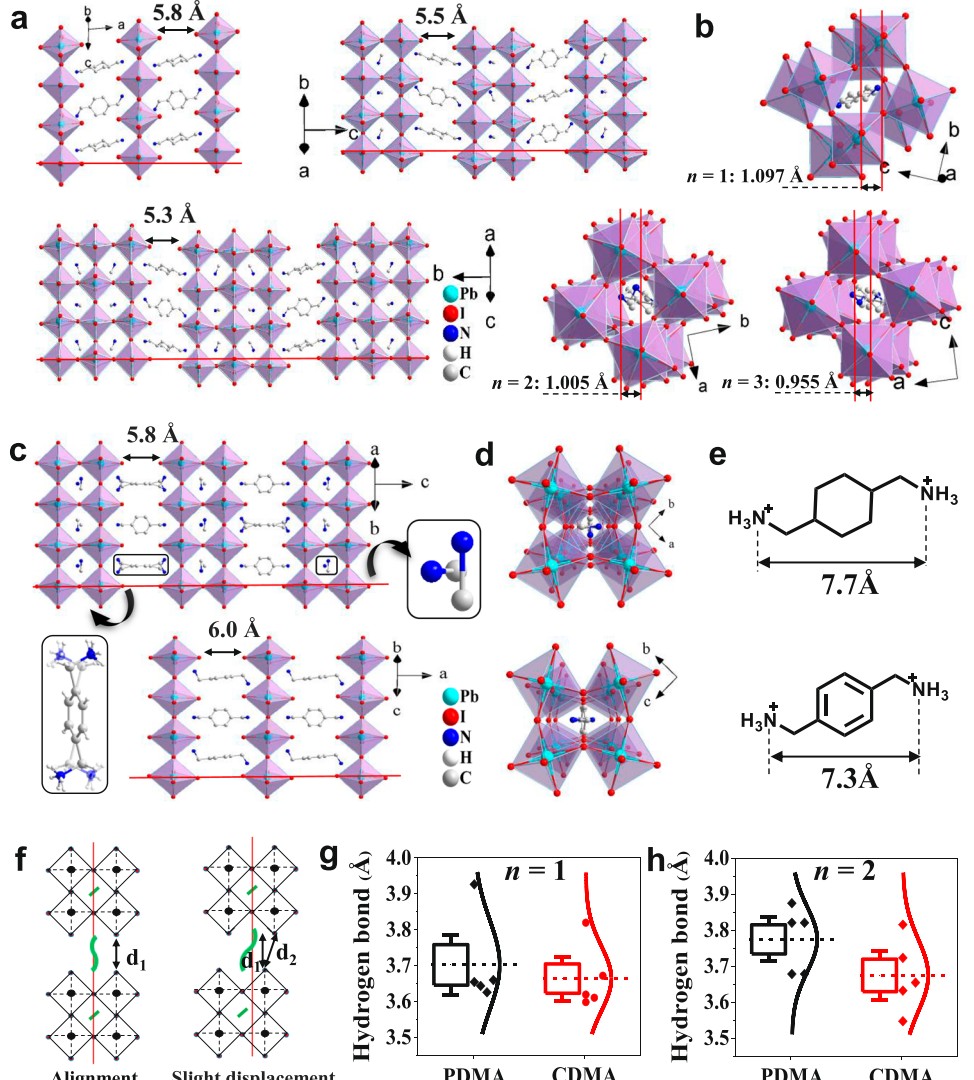

**Fig. 5 | Material design and structural analysis. a** Side-view of single-crystal structures of $(CDMA)(MA)_{n-1}Pb_nI_{3n+1}$ ($n = 1–3$). The red lines are used to highlight the displacement of adjacent inorganic slabs. **b** Top-view of single-crystal structures of $(CDMA)(MA)_{n-1}Pb_nI_{3n+1}$ ($n = 1–3$). Hydrogen atoms are omitted for clarity. The distance between the two red lines is the displacement value of the adjacent layers. **c** Side-view of single-crystal structures of $(PDMA)(MA)_{n-1}Pb_nI_{3n+1}$ ($n = 1$ and 2). **d** Top-view of single-crystal structures of $(PDMA)(MA)_{n-1}Pb_nI_{3n+1}$ ($n = 1$ and 2). Hydrogen atoms are omitted for clarity. **e** Interlayer spacer cations of CDMA and PDMA. **f** Schematic illustrated structural design of the slight interlayer-displacement DJ perovskite series. **g, h** Distributions of hydrogen-bond lengths for PDMA and CDMA series from $n = 1$ and 2, respectively. The center line represents the mean, box limits and whiskers are standard error range; the vertical curved lines and symbols represent data distribution.

designed DJ perovskite solar cells might spur new developments in scalable technology and their commercialization process. In addition, the CDMA-based DJ perovskite series also has great potential in other applications, such as 2D/3D perovskite photovoltaic technology, light-emitting diodes, photodetectors, crystallography, theoretical research, etc.

## Methods

### Materials

1,4-Cyclohexanedimethanammonium iodide (CDMA, 99.9%), 1,4-phenylenedimethanammonium (PDMA, 99.9%), methylammonium iodide (MAI, 99.9%), lead iodide ($PbI_2$, 99.99%), and poly(bis(4-phenyl)(2,4,6-trimethylphenyl)amine) (PTAA, 99.9%) were purchased from Xi'an Polymer Light Technology. Methylamine acetate (MAAc, 99.5%) was purchased Ningbo Cikang photoelectric technology Co., LTD. Lead oxide (PbO, 99.9%) was purchased from Aladdin. Hydroiodic acid (≥47.0% and ≤1.5% $H_3PO_2$ in water) was purchased from Macklin. Fullerene (C60, 99.5%) and bathocuproine (BCP, 99%) were ordered from

Lumtec. DMF (99.9%) and DMSO (99.9%) were used as received from Sigma-Aldrich. The $SnO_2$ [tin(iv) oxide, 15% in $H_2O$ colloid precursor was obtained from Alfa Aesar O colloidal dispersion], and the particles were diluted by $H_2O$ to 2.67% before use. All reagents and solvents were used directly if not specified.

### Synthesis of the DJ perovskites

For the DJ perovskite series (1,4-cyclohexanedimethanammonium) $(MA)_{n-1}Pb_nI_{3n+1}$($n = 1–3$), Stoichiometric PbO powder and organic cations were dissolved in hydrochloric acid by heating under stirring until the solution became clear. Plate-like crystal precipitates were then obtained upon cooling to room temperature.

### Preparation of perovskite precursor solutions

The 0.9 M (1,4-cyclohexanedimethanammonium)$(MA)_4Pb_5I_{16}$ (CDMA) and (1,4-phenylenedimethanammonium)$(MA)_4Pb_5I_{16}$ (PDMA) perovskite precursors was prepared by dissolving organic spacer iodide, MAI, $PbI_2$ with a molar ratio of 1:4:5 in 0.97 mL DMF and 0.03 mL DMSO

mixture solvent, and stirred in a $N_2$-purged glovebox at room temperature overnight. 1.1 M $MAPbI_3$ perovskite powder were futher dissolved in the MAAc sulution. Finally, 25% $MAAc\text{-}MAPbI_3$ perovskite solution was added into the perovskite precursors for perparing the final perovskite solution for solar cells. The PTAA solution was prepared by dissolving 1.5 mg of PTAA in 1 mL of toluene.

## Device fabrication

Glass substrates pre-patterned with the ITO (indium-tin-oxide) anode were sequentially cleaned with detergent solution, acetone, and isopropanol in a sonication bath, followed by 15 min ultraviolet ozone treatment. For the hole transporting layer, the PTAA solution predissolved in toluene was spun-coated onto the ITO substrate at 6000 rpm for 30 s and annealed at 100 °C for 10 min in $N_2$. For the perovskite layer, a trace droplet of perovskite ink (5 μL) was dripped on the PTAA-containing substrate and swiped linearly by a film applicator at the speed of 40 mm/s under an atmospheric environment. The gap between the film applicator and substrate was set as 150 μm. The temperature of the substrate is 75 °C. Then, the films were annealed at 100 °C for 10 min under an atmospheric environment. After that, 30 nm of C60 and 8 nm of BCP were sequentially deposited on the perovskite films by using a thermal evaporator under vacuum ($\approx 10^{-6}$ mbar). Finally, the Ag anode was also thermally evaporated on the BCP layer through shadow masks (vacuum $\approx 10^{-6}$ mbar). The effective device area was 0.04 cm$^2$ for the perovskite solar cells, defined by the overlapping area of the ITO and top electrodes.

## Crystal structure determination

The selected crystals were coated with oil, mounted on a loop, and transferred to a Bruker D8 Venture diffractometer equipped with a Bruker APEX-II CCD detector. Frames were collected using ω and ψ scans with 18-keV synchrotron radiation ($\lambda = 0.71073$ Å). Unit-cell parameters were refined against all data. The crystal did not show significant decay during data collection. The space-group assignment was based on systematic absences, statistics, agreement factors for equivalent reflections, and successful refinement of the structure. The structure was solved by direct methods, expanded through successive difference Fourier maps using the Olex 2, and refined against all data using the SHELXT software package[44,45]. Weighted R factors, Rw, and all goodness-of-fit indicators are based on F2. Twin laws were determined using Platon. Hydrogen atoms were inserted at idealized positions and refined using a riding model with an isotropic thermal parameter 1.2 times that of the attached carbon or nitrogen atom for $CH_2$ and $NH_3$ groups or 1.5 times that of the attached carbon atom for $CH_3$ groups. Details regarding the data quality and a summary of the residual values of the refinements are listed in cif file.

## Theoretical calculation

Calculations of formation energies were carried out by using the Vienna ab initio Simulation package[46] within the generalized gradient approximation of the Perdew−Burke−Ernzerhof (PBE) function[47]. A $9 \times 2 \times 3$, $3 \times 4 \times 4$ and $8 \times 5 \times 3$ Γ-centered k-point mesh were used for the $(PDMA)/(CDMA)Pb_2I_6 \times 2H_2O$, $(PDMA)/(CDMA)PbI_4$ and $(PDMA)/(CDMA)I_2$, respectively. These materials were fully relaxed until the force on each of the relaxed atoms was smaller than 0.02 eV Å$^{-1}$. The Grimme-D3 method was used to describe the van der Waals interaction[48].

## Characterization

**The measurements of voltage dependent current density characteristics.** The *J*−*V* curves of perovskite solar cells were measured under illumination by an Air Mass 1.5 Global (AM 1.5) solar simulator (Enli Technology Co., Ltd.) with an irradiation intensity of 100 mW cm$^{-2}$. A 20 mm × 20 mm single-crystal Si diode was used to calibrate the irradiation intensity from the solar simulator.

**The measurements of EQE.** EQE spectra were measured by a Solar Cell Spectral Response Measurement System, QE-R3011 (Enlitech Inc., Taiwan). The light intensity at each wavelength was calibrated by a single-crystal Si reference photovoltaic cell.

**Transient absorption.** Femtosecond transient absorption spectra were recorded using a commercial HELIOS ultrafast system. 800 nm pulses from a coherent Astrella regenerative amplifier (150 fs, 1 kHz, 2.0 μJ/cm$^2$) were used to pump an optical parametric amplifier (Coherent, Opera Solo) to produce 400 nm excitation pulses. As for the probe beam, it is a small fraction of the 800 nm output from the Astrella and then fed to a sapphire crystal in the HELIOS system for producing the time-delayed white light continuum. The wavelength range of the detector was set from 485 to 750 nm. All experiments were carried out at room temperature.

**Absorbance spectra.** The absorbance spectra were obtained by using a UV-vis spectrometer (Lambda 950 UV-vis spectrophotometer).

**PL spectra and time-resolved PL decay spectra.** The steady state PL spectra were measured by a HORIBA Fluorolog-III spectrofluoroemeter. The PL lifetime was probed by a customized spectrometer system (Fluorolog-III, HORIBA Jobin-Yvon Inc.) equipped with a diode laser ($\lambda = 510$ nm) for excitation.

**Scanning electron microscope (SEM) characterization.** SEM images were obtained by using a field emission scanning electron microscope microscope (SU8220, Hitachi, Japan).

**Grazing-incidence wide-angle X-ray scattering (GIWAXS) characterization.** GIWAXS patterns were obtained by using a Xenocs Xeuss SAXS/WAXS beamline system based on an X-ray wavelength of 0.6887 Å.

**Grazing incident X-ray diffraction (GIXRD) characterization.** Depth-resolved GIXRD was characterized using a Rigaku Smart Lab X-ray diffractometer at 45 kV and 200 mA, equipped with Cu Kα radiation ($\lambda = 1.54050$ Å), parallel beam optics and a secondary graphite monochromator. Before the test, the X-ray diffraction on well-recrystallized LaB6 powders was used for subtle alignment of the instrument, the acceptable LaB6 peak shift is less than 0.01° in 2θ compared to its JCPDF file. For the residual stress tests, a slow scan rate of 0.12° min$^{-1}$ was carried out to ensure fine structural information. Abbreviations of ψ, ω and φ represent instrument tilt angle, grazing incident angle, and diffraction geometry angle, respectively.

**X-ray diffraction (XRD) patterns.** XRD spectra were recorded by an X-ray diffractometer (Bruker D8 Advance) based on an X-ray wavelength of 1.5406 Å.

**Contact-angle measurements.** The contact angle was recorded by a DSA100E instrument.

**Measurements of Transient photocurrent (TPC) and Transient photovoltaic (TPV).** TPC and TPV were measured by applying a 488 nm solid-state laser (Coherent OBIS CORE 488LS) with a pulse width of ~30 ns. The current traces were recorded by a mixed-domain oscilloscope (Tektronix MDO3032) by converting the registered voltage drop across a 2 Ω resistor load connected in series to the solar cell.

**Stability test.** The thermal stability of the non-encapsulated solar cell was tested in a nitrogen atmosphere by tracking the device performance at about 85 °C. Fractional devices were aged under open-circuit conditions, and were taken out from the chamber and tested at

different time intervals. Perovskite solar cells were measured under illumination by an Air Mass 1.5 Global (AM 1.5) solar simulator (Enli Technology Co., Ltd.) with an irradiation intensity of $100\,mW\,cm^{-2}$. When measuring the cells during aging, we removed them from the aging chamber and allowed them to cool to room temperature, which typically took a few minutes. For the humidity stability, the non-encapsulated perovskite films were stored in a constant-temperature humidity chamber, taken out from the chamber, and tested at different time intervals. The relative humidity in the chamber was within $90 \pm 3\%$ during the entire aging test. Operational stability of unencapsulated devices under a white light-emitting diode lamp at maximum power point in a nitrogen-filled glove box. The measured time interval was two hours.

### Reporting summary

Further information on research design is available in the Nature Portfolio Reporting Summary linked to this article.

## Data availability

The data supporting the findings of this study are available within the paper, Supplementary Information files, and Source data. Crystal-lographic data for the structures reported in this Article have been deposited at the Cambridge Crystallographic Data Centre, under deposition numbers CCDC 2357879-2357881. These data can be obtained free of charge from The Cambridge Crystallographic Data Centre via www.ccdc.cam.ac.uk/data_request/cif. Source data are provided with this paper.

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

## Acknowledgements

Huiqiong Z. acknowledge the funding of the CAS-CSIRO Partnership Joint Project (163GJHZ2022030MI), the National Key Research and Development Program of China (2022YFB3805203), the National Natural Science Foundation of China (No. 52273245), and the Strategic Priority Research Program of Chinese Academy of Sciences (No. XDB36000000). W.Z. acknowledge the funding of the Hunan Natural Science Foundation Outstanding Youth Fund project (2024JJ4034) and the University of South China high-level talent funds (5524GC002).

## Author contributions

W.Z. and Huiqiong Z. conceived the idea and designed the experiments. W.Z. grew and refined the perovskite single crystals, provided samples for all measurements, measured the steady-state PL and ultraviolet–visible, measured the scanning electron microscope, fabricated and analyzed the perovskite devices, and wrote the manuscript. Z.L., L.Z., and H.W. did the theoretical studies. C.J. and X.W. conducted and analyzed the TA measurement. X.L. supervised the TA measurement. C.L, W.Z., and R.Y. measured and analyzed powder X-ray diffraction. S.Y. assisted in measuring fourier transform infrared spectroscopy. J.Z. measured grazing-incidence wide-angle X-ray scattering. GIWAXS was further analyzed by J.Z., Huiqiong Z. and W.Z. W.Z., Hong Z., Y.Z., and Huiqiong Z. helped with the manuscript revision and supervised the work. All authors discussed the results and commented on the manuscript.

## Competing interests

The authors declare no competing interests.
