## [Peer Review File · Nature Communications]

Ultrastable and efficient slight-interlayer-displacement 2D
Dion-Jacobson perovskite solar cellsREVIEWER COMMENTS

Reviewer #1 (Remarks to the Author):

The main finding of this manuscript is that the CDMA lacking the π - π interactions in comparison with PDMA could provide better stability of the resulting DJ perovskite and higher solar-cell performance. This is an interesting phenomenon and should be studied in depth. However, the current manuscript still contains a lot of unexplained and unclear data that needs to be further clarified. Therefore, I suggest that this article should be major revised before considering acceptance. Some the questions and suggestions below are for reference.

1. The authors used the intensity ratios of $I(202)/I(111)$ from XRD data (Supplementary Figure 17) to define the vertical orientation of 2D inorganic slabs, but the GIWAXS patterns (Figure 2c and 2d) show multiple preferential orientations of the inorganic perovskite crystals.
2. Although the resolution of the images is poor, Figures 2c and 2d seem to show at least four orientations of crystals in the CDMA-based film and three orientations in the PDMA film. The azimuthal distribution of the diffractions in Figure 2c and 2d should be shown in text. It may help to clarify whether or not distribution of the multiple preferential orientations could result in surface hydrophobicity. This is an issue worthy of in-depth study.
3. All (hkl) diffractions in GIWAXS pattern and XRD should be identified in Figure 2c and 2d. A literature report (J. Am. Chem. Soc. 2022, 144, 32, 14897) is for your reference.
4. What are ψ , ω and ϕ abbreviations for?
5. It is difficult to understand what Figure 4d and 4e are trying to express in relation to the residual strain.
6. Water resistant should be more related to the surface structure (see the SEM images in Figure 3a and 3b) than the residual strain inside the crystals. Notably, the preferred orientation of PDMA- and CDMA-based perovskite crystals are different as shown in Figure 2c and 2d. It should be confirmed whether the surface hydrophilicity is changed due to its different orientation of crystals. The current research data of crystalline structures cannot reasonably explain why CDMA-based film can have excellent water resistance. Notably, CDMA without the π - π interactions does not necessarily stack along a single direction. Is it possible that an arrangement different from that of the crystals forms a hydrophobic layer on the film surface? Some experiments should be tried to directly obtain information about surface structure such as GISAXS measurement with varied tilted angles, SEM image of cross-sectional film or depth analysis of X-ray photoelectron spectroscopy (XPS).
7. In Figure 2c and 2d, no diffraction peaks from low-n are observed in GIWAXS and XRD data, although the TA spectra have shown low-n phase. Does this mean that both the films are major 3D-like phases and highly-disordered low-n phases?

8. The depth-dependent strain distribution measurement for polycrystalline film mainly provides residual stress from macroscopic film compression or expansion, and does not directly give stress within a single grain or crystal. In Figure 4a and 4b, the peaks largely broaden with increasing ψ . Thus data quality doesn't enough to give the right message, because the strong preferential orientations could affect the result. It is a suggestion that grain size and crystallinity analysis can be utilized to distinguish the differences between the two films.

9. Why peak positions of diffractions are different between in-plane and out-of-plane directions in Supplementary Fig. 18.

10. GIXRD and GIWAXS names can be used uniformly in the manuscript.

11. What is wavelength of XRD? It should be revealed in experimental section.

12. The single crystal structure does not seem to explain why CDMA can have good water resistance.

Reviewer #2 (Remarks to the Author):

Zhang and coworkers present a study of perovskite solar cells employing a novel large organic cation named CDMA. Such cation, when alone, forms a 2D Dion-Jacobson perovskite structure, while when incorporated in a standard 3D perovskite improves solar cell performance and stability.

Such improvement is attributed to the fact that the large cations may passivate defects at interfaces, grain boundaries and in the bulk.

The strategy of incorporating large cations in perovskite solar cells has been successfully tested in many other studies and is convincingly beneficial. On one side this means that the topic is extremely relevant for the community and timely. On the other hand, the extensive previous use of the same strategy chips at the novelty and thus it is important to understand how the current work stands out. The authors present impressive performances and extensive characterization. However, it seems to me that, at the current stage of perovskite solar cell research, a more in-depth understanding of the effect of the large cations is needed. In particular, I feel several pieces of information provided by the authors do not match to each other and need clarification. In particular, the following points need to be addressed:

- Why the efficiency and stability is compared with MAPI and not FAPI or the multi-cation perovskites that offer significant improvements?

- The interpretation of time-resolved PL decays does not seem consistent. The decay time is almost invariably due to trap-assisted recombination, so if it becomes faster, as it does when CDMA is added to the perovskite, it means that trap-assisted recombination becomes faster. The authors claim the opposite.

- Still concerning photophysics interpretation of PL decays: the observed long decay time is most probably not radiative. No physical reason is provided why the initial decay should be to traps, then

at long times traps are not effective anymore and only radiative decay is present. This not in accordance to any accepted rate equation or known decay process. In any case, radiative recombination is bimolecular and thus its characteristic time should decrease linearly with increasing excitation fluence.

- Concerning the absorption spectra: why the $n=1$ DJ perovskite has no exciton peak in absorption? It should display the best-resolved and sharpest exciton peak of the whole DJ series.

- Still on absorption spectra: are oscillations related only to the formation of 2D perovskite phases or there is a contribution from intrinsic confinement – see recent work by Laura Herz and coworkers ACS Energy Lett. 2023, 8, 2543–2551. Does the amplitude of the oscillations match the expected fraction of 2D perovskite phases? Does it change with the amount of large cations included?

- Concerning the formation of various 2D phases: the authors claim when discussing transient absorption spectra that when CDMA is added only phases with $n=3$ or larger form, why no trace of $n=1$ and $n=2$ phase is detected. How is it possible? Why the various phases do not form according to simple statistics? Other films in different figures show that $n=1$ and $n=2$ phases do form with significant abundance.

- Concerning level alignment: if the bandgap energy determined by TAUC plot, what is the effect of the exciton binding energy? Its magnitude is several hundreds of meV, thus it could significantly change the whole picture and could not be neglected at all.

Overall, the manuscript showcases notable advancements in performance, and the authors deserve recognition for their thorough characterization efforts. Nonetheless, in order for the manuscript to be considered for publication in Nature Communications, I recommend that the underlying microscopic mechanisms driving the favorable outcomes of the large cations is explained. After undergoing a comprehensive review process, the manuscript could potentially be suitable for publication.

Response Letter to Reviewers

(Text in black: comments and questions of reviewers; Text in blue: our response to reviewers; Text with yellow highlight in manuscript and SI: our revision; Figure R1: figures from Response letter; Supplementary Fig. 1: figures from Supplementary information; Fig. 1: figures from main text; And superscript ¹: references from main text. The "1" here represents for any number.)

Reviewer #1 (Remarks to the Author):

The main finding of this manuscript is that the CDMA lacking the π - π interactions in comparison with PDMA could provide better stability of the resulting DJ perovskite and higher solar-cell performance. This is an interesting phenomenon and should be studied in depth. However, the current manuscript still contains a lot of unexplained and unclear data that needs to be further clarified. Therefore, I suggest that this article should be major revised before considering acceptance. Some the questions and suggestions below are for reference.

1. The authors used the intensity ratios of $I(202)/I(111)$ from XRD data (Supplementary Figure 17) to define the vertical orientation of 2D inorganic slabs, but the GIWAXS patterns (Figure 2c and 2d) show multiple preferential orientations of the inorganic perovskite crystals.

✓ Response:

Thanks a lot for the reviewer's valuable question. Please note, all the mentioned Figure 2c and 2d on the GIWAXS measurements in the questions are mislabeled for the figure number, and they should correspond to Figure 3c and 3d in the main text.

In the revised manuscript, we discussed the FWHM of XRD peaks to evaluate the crystallinity of the perovskite films and the orientation of the inorganic perovskite crystals have been explored mainly by GIWAXS patterns. Please see the main text for details: The first paragraph of "Characterization of the Perovskite Films" on page 6 in the revised manuscript.

2. Although the resolution of the images is poor, Figures 2c and 2d seem to show at least four orientations of crystals in the CDMA-based film and three orientations in the PDMA film. The azimuthal distribution of the diffractions in Figure 2c and 2d should be shown in text. It may help to clarify whether or not distribution of the multiple preferential orientations could result in surface hydrophobicity. This is an issue worthy of in-depth study.

✓ Response:

Thanks for the suggestions. We have analyzed the azimuthal distribution of the diffractions [(202) and (111)] in GIWAXS patterns of PDMA and CDMA based films (Figure R1). It is shown that both (111) and (202) diffractions of CDMA film are strong at 45° , and PDMA film shows a strong (111) peak at 90° . The orientations of PDMA and CDMA do exhibit some differences. While about the surface hydrophobicity, we'd like to point out that there is a small (011) peak at around 8.3° of 1D (PDMA) $\text{Pb}_2\text{I}_6 \cdot 2\text{H}_2\text{O}$ in the XRD pattern of fresh PDMA based perovskite film (please see Figure R2c), and this peak becomes prominent when the PDMA based film exposed under relative humidity (RH) of

85–90% for 2 h (please see Fig 4d in the manuscript). This phenomenon originates from that the PDMA-based perovskites easily form a 1D hydrate (PDMA)Pb₂I₆·2H₂O by reacting with water molecules (*ACS Energy Lett.* 2021, 6, 337–344):

Contrarily, there are no peak of this hydrate in XRD patterns for both fresh and aged CDMA based perovskite films (please see Figure R2c and Fig 4e in the manuscript). Thus, we consider that the reaction between PDMA-based perovskites and water molecules could be the dominated reason for the more hydrophilic of PDMA based perovskite films.

Figure R1. Azimuthal scans of the (202) and (111) diffraction from GIWAXS measurements.

3. All (hkl) diffractions in GIWAXS pattern and XRD should be identified in Figure 2c and 2d. A literature report (*J. Am. Chem. Soc.* 2022, 144, 32, 14897) is for your reference.

✓ **Response:**

Thanks for the valuable suggestion. Accordingly, we have identified (hkl) diffractions in GIWAXS and XRD patterns based on several literature reports (Small 2020, 16, 2003098; *J. Am. Chem. Soc.*, 2020, 142, 11114-11122; *Adv. Mater.* 2021, 33, 2105083; *Nature*, 2016, 536, 312-316; *J. Am. Chem. Soc.* 2015, 137, 7843-7850; et. al.). Details are shown in Fig 3c, 3d in the revised manuscript (Figure. R2a, R2b) and the revised Supplementary Fig. 17a (Figure. R2c).

Figure R2. **a, b**, GIWAXS images of the DJ perovskite films. **c**, XRD image of the DJ perovskite films.

4. What are ψ , ω and ϕ abbreviations for?

✓ **Response:**

We really appreciate the reviewer's careful review and valuable comments. Abbreviations of ψ , ω and ϕ represent instrument tilt angle, grazing incident angle, and diffraction geometry angle, respectively

(Figure R3). Descriptions of these abbreviations have been added to the characterization section [Grazing incident X-ray diffraction (GIXRD) characterization, Page 14 in the revised manuscript].

Figure R3. The relationship between the instrumental angles.

5. It is difficult to understand what Figure 4d and 4e are trying to express in relation to the residual strain.

✓ Response:

Thanks for the comment. We agree that it's difficult to demonstrate the complex issue about residual strain in simplified schemes. The two schematic diagrams also do not contribute much to this work and have been removed in the revised manuscript.

6. Water resistant should be more related to the surface structure (see the SEM images in Figure 3a and 3b) than the residual strain inside the crystals. Notably, the preferred orientation of PDMA- and CDMA-based perovskite crystals are different as shown in Figure 2c and 2d. It should be confirmed whether the surface hydrophilicity is changed due to its different orientation of crystals. The current research data of crystalline structures cannot reasonably explain why CDMA-based film can have excellent water resistance. Notably, CDMA without the π - π interactions does not necessarily stack along a single direction. Is it possible that an arrangement different from that of the crystals forms a hydrophobic layer on the film surface? Some experiments should be tried to directly obtain information about surface structure such as GISAXS measurement with varied titled angles, SEM image of cross-sectional film or depth analysis of X-ray photoelectron spectroscopy (XPS).

✓ Response:

Thanks for the reviewer's valuable comments. According to the reviewer's suggestion, we further investigated the cross-sectional scanning electron microscope (SEM). Images of the cross-sectional SEM show no obvious difference between PDMA- and CDMA-based perovskite films (Figure R4). We agree that the water resistant could be related to the surface structure, the orientation of crystals or element distribution et. al., while in this work, we consider that the reaction between PDMA-based perovskites and water molecules could be the dominated reason for the more hydrophilic of PDMA based perovskite films as discussed in question 2.

Figure R4. Cross-sectional SEM images of the DJ perovskite films.

7. In Figure 2c and 2d, no diffraction peaks from low- n are observed in GIWAXS and XRD data, although the TA spectra have shown low- n phase. Does this mean that both the films are major 3D-like phases and highly-disordered low- n phases?

✓ **Response:**

Thanks for the reviewer's question. According to the results of TA spectra excited from the backside and frontside, both films show a graded phase distribution. Near the surface of the film is mainly composed of high n -value perovskite phases (3D-like) and the composition of low n -value 2D perovskite components are significantly increased in the region near the PTAA layer. As shown in Supplementary Fig. 23 in the SI, the $n > 3$ compounds show mainly the (111) and (202) reflections (here $n=5$ for the active layer in solar cell devices), clearly indicating the well orientation of the perovskite compounds. This phenomenon also has been reported in literatures (Adv. Mater. 2021, 33, 2105083; J. Am. Chem. Soc. 2015, 137, 7843-7850; et. al.).

8. The depth-dependent strain distribution measurement for polycrystalline film mainly provides residual stress from macroscopic film compression or expansion, and does not directly give stress within a single grain or crystal. In Figure 4a and 4b, the peaks largely broaden with increasing ψ . Thus data quality doesn't enough to give the right message, because the strong preferential orientations could affect the result. It is a suggestion that grain size and crystallinity analysis can be utilized to distinguish the differences between the two films.

✓ **Response:**

Thanks for the reviewer's suggestion. We agree with the reviewer's opinion that the depth-dependent strain distribution measurement for polycrystalline film mainly provides residual stress from macroscopic film compression or expansion, and does not directly give stress within a single grain or crystal. It is reported that the release of the residual stress from macroscopic film compression or expansion can lead to better stability under external stimulate (Nat. Commun. 2019, 10, 815; Angew. Chem.Int. Ed. 2022, 61, e2022082). We further analyzed the data in Figure 4a and 4b. It shows a slight fluctuation in the Full Width at Half Maximum of the peaks with increasing ψ as shown in Figure R5a, which can lead to reliable results. Besides, another method based on XRD data, Williamson–Hall plots were further applied to calculate the residual strain in perovskite films using Equation: $\beta \cos\theta = \varepsilon(4\sin\theta) + k\lambda/D$, where β is total broadening of XRD peaks, defined as FWHM, and θ is diffraction angle, ε is residual strain, K is Scherrer constant (≈ 0.9 for perovskite), λ is wavelength of X-ray (1.5406 Å), and D is crystal size of perovskite film. The detailed linear fitting and data are shown in Figure R5b. The fit value of the calculated residual strain further shows an almost free residual strain of the CDMA films, consistent with the depth-dependent GIXRD measurements.

Figure R5. **a**, Linear fit of FWHM- $\sin^2\phi$ in the different regions for the PDMA- and CDMA-based film. **b**, Williamson–Hall plots fitting of DJ perovskite films. **c**, Comparison of FWHM of the (111)-oriented XRD peaks.

Accordingly, we have further refined the grain size and crystallinity analysis to distinguish the differences between the two films. As shown in Fig. 3a and 3b in the manuscript, the SEM image of the PDMA-based perovskite film shows a uniform grain distribution with small sizes, and the CDMA-based film exhibits a distinct enlargement of grain sizes. In contrast to the PDMA-based films, CDMA exhibit a relatively narrower FWHM and an increased intensity of diffraction peaks in XRD pattern, indicating the improved crystallinity and the enlarged grain size according to Scherrer's equation (Figure R2 and R5c).

Also, Figures R5b and R5c are added to Supplementary Information (Supplementary Fig. 21c and Fig. 17b) with brief descriptions, respectively.

9. Why peak positions of diffractions are different between in-plane and out-of-plane directions in Supplementary Fig. 18.

✓ **Response:**

We really appreciate the reviewer's careful review. In the drawing process, we accidentally used the X column of in-plane for drawing the out-of-plane figure. We checked the data and revised them as followed Figure R6 and also shown in Supplementary Fig. 18.

Figure. R6. Comparison of line-cut profile curves of 2D GIWAXS along in-plane directions and out-of-plane.

10. GIXRD and GIWAXS names can be used uniformly in the manuscript.

✓ **Response:**

Thanks for the reviewer's suggestion. In this work, GIXRD and GIWAXS are derived from different measurements with different instruments. Depth resolved GIXRD were characterized using a Rigaku

Smart Lab X-ray diffractometer at 45 kV and 200 mA, equipped with Cu K α radiation ($\lambda = 1.54050 \text{ \AA}$), parallel beam optics and a secondary graphite monochromator. GIWAXS patterns were obtained by using a Xenocs Xeuss SAXS/WAXS beamline system based on an X-ray wavelength of 0.6887 \AA . Therefore, we thought it would be better to name them separately to prevent confusion.

11. What is wavelength of XRD? It should be revealed in experimental section.

✓ **Response:**

Thanks for the careful review. The wavelength of XRD is 1.5406 \AA . We have updated it in experimental section in the revised manuscript.

12. The single crystal structure does not seem to explain why CDMA can have good water resistance.

✓ **Response:**

Thanks for the reviewer's valuable comments. DJ types with ditopic diammonium cations are deemed to strengthen the connection between inorganic layers, thereby enhancing stability. Among them, large spacer cations connect adjacent inorganic perovskite slabs via hydrogen bonding, which has been recognized as a key to improving structural stability for water resistance (*J. Am. Chem. Soc.* 2018, 140, 12226; *J. Am. Chem. Soc.* 2021, 143, 19901.). Therefore, we extracted the data on hydrogen bonds between space cations and inorganic slabs of $n = 1$ and $n = 2$ 2D perovskites based on the obtained CDMA and PDMA crystal structures, as shown in Supplementary Fig. 36 and Fig. 5e. It shows that the average hydrogen bond lengths of both $n = 1$ and $n = 2$ PDMA-based perovskites are larger than those of CDMA-based crystals, suggesting a relatively poor interaction in relation to structural stability. Moreover, the equatorial Pb-I-Pb angles that interacted with the PDMA cations show smaller than the CDMA-based perovskites (Supplementary Fig. 35b), indicating notable twist features and possible effect on structural stability. The weak hydrogen-bond interaction and relatively twisted equatorial Pb-I-Pb angles could be the reason that the PDMA-based perovskites easily form a 1D hydrate (PDMA) $\text{Pb}_2\text{I}_6 \cdot 2\text{H}_2\text{O}$ by reacting with water molecules (*ACS Energy Lett.* 2020, 6, 337). In contrast, the relatively strong hydrogen-bond interaction of CDMA cations can strengthen the connection between inorganic layers, thereby resisting the attack of water molecules, protecting the inorganic perovskite layers, and significantly improving stability. We have added this part of the discussion as follows (Last paragraph of Structural analyses in Page 10 in the revised manuscript):

In addition to the slightly decrease layer distance, we further investigated the microscopic mechanism facilitating the favorable stability by the large cations. For the 2D perovskites, large spacer cations connect adjacent inorganic perovskite slabs via hydrogen bonding, which has been recognized as the key to improving structural stability. Therefore, we extracted the data on hydrogen bonds between space cations and inorganic slabs of $n = 1$ and $n = 2$ 2D perovskites based on the obtained CDMA and PDMA crystal structures, as shown in Supplementary Fig. 36. Statistical comparisons of their corresponding distributions were depicted in Fig. 5e. It shows that the average hydrogen bond lengths of both $n = 1$ and $n = 2$ PDMA-based perovskites are larger than those of CDMA-based materials, suggesting a relatively poor interaction in relation to structural stability. Moreover, the equatorial Pb-I-Pb angles that interacted with the PDMA cations show smaller than the CDMA-based perovskites (Supplementary Fig. 35b), indicating notable twist features and possible seriously affect structural stability. The weak hydrogen-bond interaction and relatively twisted equatorial Pb-I-Pb angles could be the reason that the

PDMA-based perovskites easily form a 1D hydrate (PDMA)Pb₂I₆·2H₂O by reacting with water molecules.³¹ In contrast, the relatively strong hydrogen-bond interaction of CDMA cations can strengthen the connection between inorganic layers, thereby resisting the attack of water molecules, protecting the inorganic perovskite layers, and significantly improving stability.

Reviewer #2 (Remarks to the Author):

Zhang and coworkers present a study of perovskite solar cells employing a novel large organic cation named CDMA. Such cation, when alone, forms a 2D Dion-Jacobson perovskite structure, while when incorporated in a standard 3D perovskite improves solar cell performance and stability. Such improvement is attributed to the fact that the large cations may passivate defects at interfaces, grain boundaries and in the bulk. The strategy of incorporating large cations in perovskite solar cells has been successfully tested in many other studies and is convincingly beneficial. On one side this means that the topic is extremely relevant for the community and timely. On the other hand, the extensive previous use of the same strategy chips at the novelty and thus it is important to understand how the current work stands out. The authors present impressive performances and extensive characterization. However, it seems to me that, at the current stage of perovskite solar cell research, a more in-depth understanding of the effect of the large cations is needed. In particular, I feel several pieces of information provided by the authors do not match to each other and need clarification. In particular, the following points need to be addressed:

1. Why the efficiency and stability is compared with MAPI and not FAPI or the multi-cation perovskites that offer significant improvements?

✓ **Response:**

We appreciate the positive comments and thank for the valuable suggestions. We found that it is more likely to obtain different n-value 2D perovskites by using MA cations than FA or mixed FA and MA cations in the synthesis of PDMA- and CDMA-based single crystals. Thus, the current work mainly focuses on the synthesis of MA-based 2D perovskites and their application in solar cells. Further work on FA or other multi-cations are going on in the lab and it is expected to get improvement in device performance.

2. The interpretation of time-resolved PL decays does not seem consistent. The decay time is almost invariably due to trap-assisted recombination, so if it becomes faster, as it does when CDMA is added to the perovskite, it means that trap-assisted recombination becomes faster. The authors claim the opposite.

✓ **Response:**

Thanks for the careful review. In this work, it is worthy to point out that the photoluminescence and time-resolved photoluminescence (TRPL) spectroscopy were applied to study charge extraction kinetics of the 2D perovskite on a substrate with hole transport layer (PTAA). It is different from the method by which TRPL characterization revealed the recombination by directly coating perovskite on a bare glass

substrate without transport layer. Remarkable PL and TRPL quenching are observed on the CDMA-based perovskite film, indicating efficient charge transfer from the active layer to the transport layer. This phenomenon is consistent with most reported literature, such as Adv. Funct. Mater.2023, 33, 2212606; Adv. Mater. 2018, 30, 1800710; Energy Environ. Sci., 2020, 13, 3093; etc.

3. Still concerning photophysics interpretation of PL decays: the observed long decay time is most probably not radiative. No physical reason is provided why the initial decay should be to traps, then at long times traps are not effective anymore and only radiative decay is present. This not in accordance to any accepted rate equation or known decay process. In any case, radiative recombination is bimolecular and thus its characteristic time should decrease linearly with increasing excitation fluence.

✓ **Response:**

Thanks a lot for the reviewer's question. The photoluminescence (PL) decays were also applied to study charge extraction kinetics of the 2D perovskite on a substrate with hole transport layer (PTAA). According to the reviewer's suggestion, we have revised the description on PL decays in the main text and Supplementary information. The details are as follows.

In the revised manuscript page 5 (Last paragraph of Device characteristics in the revised manuscript): *Supplementary Fig. 8 shows the PL spectra and decay of DJ perovskites on the PTAA transport layer. The fit data are summarized in Supplementary Table 4. Remarkable PL quenching is observed on the CDMA-based perovskite film, indicating efficient charge transfer from the active layer to the transport layer. In addition, it is shown that the decay times of CDMA-based perovskite films are smaller than the PDMA, which suggests that the CDMA active layer has a relatively high charge transfer ability from perovskite to PTAA layer.*^{36,37}

In the revised Supplementary information page 44 (Below Supplementary Table 4):

The time-resolved photoluminescence decay curves can be fitted by the bi-exponential function:

$$y = A_1 \exp(-t/\tau_1) + A_2 \exp(-t/\tau_2) + y_0$$

Where A_1 and A_2 are the relative amplitudes; and τ_1 and τ_2 are the decay time constants.^{57,58}

4. Concerning the absorption spectra: why the n=1 DJ perovskite has no exciton peak in absorption? It should display the best-resolved and sharpest exciton peak of the while DJ series.

✓ **Response:**

Thanks for the reviewer's question. The absorption spectra of n=1-3 DJ perovskites are measured based on single-crystal samples, as shown in Supplementary Fig. 13a. In contrast to the films, exciton peaks are smaller in the absorption spectrum of single crystals. An apparent exciton peak of n = 1 can be seen at about 500 nm, as shown in Supplementary Fig. 13a (Figure R7), which is consistent with the absorption of most reported crystal powders (e.g., J. Am. Chem. Soc. 2021, 143, 12063, J. Am. Chem. Soc. 2019, 141, 12880, etc.). We further investigated the absorption spectrum of n =1 film and did find a distinct and sharp exciton peak at about 512 nm.

Figure R7. a, Optical properties of (CDMA)(MA)_{n-1}Pb_nI_{3n+1} ($n=1-3$) perovskites crystals. b, Absorption spectrum of the $n=1$ perovskite film.

5. Still on absorption spectra: are oscillations related only to the formation of 2D perovskite phases or there is a contribution from intrinsic confinement – see recent work by Laura Herz and coworkers ACS Energy Lett. 2023, 8, 2543–2551. Does the amplitude of the oscillations match the expected fraction of 2D perovskite phases? Does it change with the amount of large cations included?

✓ **Response:**

Thanks a lot for the reviewer's question. The amplitude of the oscillations matches the expected fraction of 2D perovskite phases. The ground-state bleaching peaks of these specific wavelengths have been verified and reported by numerous publications, and the 2D phases can be identified by comparing the peak positions in these references (*Adv. Mater.* 2019, 31, 1901966; *Adv. Energy Mater.* 2021, 11, 2002733). Besides, the position of these peaks does not change with the change of the amount of cations, but the intensity changes [e.g., n-Amylammonium, *Matter* 2021,4, 582-599 ($n=4$); ACS Energy Lett. 2022, 7, 1842 ($n=5$)]. Therefore, these 2D peaks (oscillations) are related only to the formation of 2D perovskite phases.

6. Concerning the formation of various 2D phases: the authors claim when discussing transient absorption spectra that when CDMA is added only phases with $n=3$ or larger form, why no trace of $n=1$ and $n=2$ phase is detected. How is it possible? Why the various phases do not form according to simple statistics? Other films in different figures show that $n=1$ and $n=2$ phases do form with significant abundance.

✓ **Response:**

Thank you for your careful review. We have carefully examined and enlarged the TA spectra of CDMA-based films, as shown in Figure R8. It is shown that there is an $n=2$ perovskite peak (Figure R8b), but the peak intensity is extraordinarily weak and easy to ignore. According to the reviewer's comment, we further emphasized this inconspicuous 2D perovskite phase ($n=2$) in the main text. The details are as follows:

By comparison, the CDMA films have an additional high n -value 2D phase ($n=6$), and their bleaching peak intensities enhance with the increase of perovskite layers. Particularly, the peak of $n=2$ in CDMA-based perovskite films is extremely inconspicuous and almost undetected. (In the second paragraph of the Characterization of the perovskite films section on page 7, line 9)

Figure. R8. a, TA spectra under different delay times for the CDMA film under back-photoexcitation. **b,** Local enlarged view of TA spectrum.

We further examined the peak of the $n = 1$ perovskite and confirmed that it was not detected. In this work, PDMA and CDMA perovskite films for solar cells are fabricated based on nominal $n = 5$ DJ perovskite [e.g., $(\text{CDMA})(\text{MA})_4\text{Pb}_5\text{I}_{16}$]. In addition, the solvent of the perovskite precursor includes MAAc component. MA cations have a relatively high proportion in the precursor, so it is difficult to form an MA-free $n = 1$ perovskite phase.

According to the reviewer's comment, we also checked all the figures in main text and Supplementary. In addition to the characterization of pure $n = 1$ or $n = 2$ perovskite single crystals (e.g., Supplementary Fig. 13), perovskite films and solar cells fabricated based on the nominal $n = 5$ precursors can hardly be detected for the features or peaks of $n = 1$ and $n = 2$ perovskite phases, such as the UV-vis absorption spectra (Supplementary Fig. 17c).

7. Concerning level alignment: if the bandgap energy determined by TAUC plot, what is the effect of the exciton binding energy? Its magnitude is several hundreds of meV, thus it could significantly change the whole picture and could not be neglected at all.

✓ **Response:**

Thanks a lot for the reviewer's question. We agree with the reviewer about the effect of exciton binding energy on the energy level alignment of 2D perovskites. In this work, the energy diagram was deduced from Tauc plot and UPS measurement.

The exciton binding energy increases with n value and reaches its maximum at $n=1$, which could relatively significant enlarge the energy band gap of the perovskites with small n values (Adv. Energy Mater. 2022, 12, 2202333). In this case, it would not change the trend of energy-level alignment in the schematic diagram about charge transfer in the device.

On the other hand, it is very difficult to determine the value of exciton binding energy for perovskite films with different n values since the resulted perovskites (especially with large n values) are composited by mixed phases with different n values in this work.

Thus, we have revised the caption of Supplementary Fig. 16 and noted that the diagram was obtained without considering the exciton binding energy.

Overall, the manuscript showcases notable advancements in performance, and the authors deserve recognition for their thorough characterization efforts. Nonetheless, in order for the manuscript to be considered for publication in Nature Communications, I

recommend that the underlying microscopic mechanisms driving the favorable outcomes of the large cations is explained. After undergoing a comprehensive review process, the manuscript could potentially be suitable for publication.

✓ **Response:**

We really appreciate the reviewer's positive comments and valuable suggestions. Accordingly, we further underlined the microscopic mechanism facilitating the favorable outcomes by the large cations. It includes two aspects, which are the I··I distance between the inorganic layer involved in facilitating interlayer charge transport and the N-H··I hydrogen-bond interaction in relation to structural stability. Details are as follows and also shown in the revised manuscript on page 10 and 11: (Paragraph 2 and 3 of the Structural analyses section in the revised manuscript)

We further measured the size of these two cations, in which the CDMA and PDMA cations show lengths of 7.7 and 7.4 Å, respectively (Fig. 5c). In general, 2D perovskites with longer dimensional cations have larger layer spacing. However, the slight-interlayer-displacement CDMA perovskites with longer cation lengths show shorter layer spacing, slightly decreasing with the increased layer number of 2D perovskite layers (5.8, 5.5, and 5.3 Å for n = 1-3). The I··I distance between the inorganic layer for n = 1-3 perovskites are 5.89, 5.63, and 5.48 Å, respectively (Supplementary Fig. 35a). Differently, layer spaces of aligned PDMA perovskites are 6.0 Å (n = 1) and 5.8 Å (n = 2). The schematic diagram to illustrate the difference between the slight-interlayer-displacement and 0-displacement DJ perovskite series is shown in Fig. 5d. Distinctly, decreasing interlayer spaces and tuning their mutual alignment are significant parameters for controlling the optoelectronic and electrostatic properties as they strengthen electronic interactions²⁹ and strain⁴¹ between the inorganic layers, and consequently, facilitate interlayer charge transport and structural stability.

In addition to the slightly decrease layer distance, we further investigated the microscopic mechanism facilitating the favorable stability by the large cations. For the 2D perovskites, large spacer cations connect adjacent inorganic perovskite slabs via hydrogen bonding, which has been recognized as the key to improving structural stability. Therefore, we extracted the data on hydrogen bonds between space cations and inorganic slabs of n = 1 and n = 2 2D perovskites based on the obtained CDMA and PDMA crystal structures, as shown in Supplementary Fig. 36. Statistical comparisons of their corresponding distributions were depicted in Fig. 5e. It shows that the average hydrogen bond lengths of both n = 1 and n = 2 PDMA-based perovskites are larger than those of CDMA-based materials, suggesting a relatively poor interaction in relation to structural stability. Moreover, the equatorial Pb-I-Pb angles that interacted with the PDMA cations show smaller than the CDMA-based perovskites (Supplementary Fig. 35b), indicating notable twist features and possibly seriously affect structural stability. Based on these, the weak hydrogen-bond interaction and relatively twisted equatorial Pb-I-Pb angles could be the reason that the PDMA-based perovskites easily form a 1D hydrate (PDMA)₂PbI₆·2H₂O by reacting with water molecules.³¹ In contrast, the relatively strong hydrogen-bond interaction of CDMA cations can strengthen the connection between inorganic layers, thereby resisting the attack of water molecules, protecting the inorganic perovskite layers, and significantly improving stability.

REVIEWER COMMENTS

Reviewer #1 (Remarks to the Author):

The revised manuscript still has some unresolved issues and does not provide reasonable solutions. Therefore, I cannot recommend it for publication now.

1. This study should reasonably explain why (CDMA)Pb₂I₆·H₂O is not formed. The current explanation regarding hydrogen bond lengths appears to be a bit of a stretch. The most significant difference between CDMA and PDMA lies in the orientation of the 2D inorganic slabs, as illustrated in Figure R1. Should the authors redefine the correct (hkl) and consider the possibility of a flat-on orientation of the slabs covering the surface leading to hydrophobicity?

2. The definition of (hkl) does not comply with the crystal parameters of orthorhombic, as indicated in Supplementary Table 6. In Figure 3c and 3d, only (111) and (311) match to the calculation of orthorhombic ($1/d^2 = (h^2/a^2) + (k^2/b^2) + (l^2/c^2)$), but other do not. For example, q peaks of (00-2), (-202) and (-15-2) should be 0.2595, 1.54 and 3.636 Å⁻¹.

3. The authors are unable to define (hkl) correctly, leading to an inability to clarify the orientation relationship between the crystal and the substrate. In the section of Characterization of the perovskite films, the authors mentioned that “In contrast, CDMA-based film shows relatively strong and discrete Bragg spots and out-of-plane distribution.... It suggests a high degree of the preferential vertical orientation of the 2D perovskite inorganic frameworks, forming a favorable charge-transport channel in the solar cells.” Based on the orthorhombic lattice (Supplementary Table 6), higher intensity of (111) does not indicate the preferential vertical orientation.

4. Figure S18 shows that CDMA has higher intensity of (111) diffraction than that of PDMA at the out-of-plane direction. However, Figure R1 presents a result opposite to those in Figure S18.

5. It should added FTIR data to demonstrate the hydrogen bond

6. In text, “The ultraviolet-visible (UV-Vis) absorption... DJ perovskite films (Supplementary Fig. 17b)”. The Fig. 17b should be corrected as Fig. 17c.

7. Some X-ray data use q but some use 2θ, it is recommended to use q uniformly.

Reviewer #2 (Remarks to the Author):

The revised manuscript by Zhang and coworkers addresses the weak points that have been raised in the review process. The authors have replied convincingly to each of the criticisms and the manuscript, in my opinion, is significantly improved.

I recommend proceeding to publication.

Response Letter to Reviewers

(Text in black: comments and questions of reviewers; Text in blue: our response to reviewers; Text with yellow highlight in manuscript and SI: our revision; Figure R1: figures from Response letter; Supplementary Fig. 1: figures from Supplementary information; Fig. 1: figures from main text; The "1" here represents for any number.)

Reviewer #1 (Remarks to the Author):

The revised manuscript still has some unresolved issues and does not provide reasonable solutions. Therefore, I cannot recommend it for publication now.

1. This study should reasonably explain why $(\text{CDMA})\text{Pb}_2\text{I}_6 \cdot \text{H}_2\text{O}$ is not formed. The current explanation regarding hydrogen bond lengths appears to be a bit of a stretch. The most significant difference between CDMA and PDMA lies in the orientation of the 2D inorganic slabs, as illustrated in Figure R1. Should the authors redefine the correct (hkl) and consider the possibility of a flat-on orientation of the slabs covering the surface leading to hydrophobicity?

✓ Response:

Thanks a lot for the reviewer's valuable question. The hydrophobicity of low-dimensional perovskite materials is generally related to organic components. The strength of interaction between organic spacers and inorganic slabs is an intrinsic factor that determines the material stability; thus, we consider that it is reasonable to use hydrogen bond data to explain their difference in stability (Ref. 43 in main text).

Fig. R1: DFT calculations. a, DFT calculated formation energy of the PDMA-based materials. b, DFT calculated formation energy of the CDMA-based materials.

Additionally, DFT calculations were further performed to reasonably explain that $(\text{CDMA})\text{Pb}_2\text{I}_6 \cdot \text{H}_2\text{O}$ is not formed, as shown in Figure R1. For the PDMA-based perovskite, $\Delta E_{\text{PDMA}} = E_{\text{Product}} - E_{\text{Reactant}} = -0.4 \text{ eV}$, where the total energy of the reactant is higher than the total energy of the product, indicating that

(PDMA)PbI₄ can spontaneously react with water to synthesize (PDMA)Pb₂I₆·2H₂O and (PDMA)I₂ with lower total energy. By contrast, the CDMA-based results show the total energy of the reactant is lower than the total energy of the product ($\Delta E_{\text{CDMA}} = E_{\text{Product}} - E_{\text{Reactant}} = 0.472 \text{ eV}$), indicating that (CDMA)PbI₄ cannot react spontaneously with water and is difficult to synthesize (CDMA)Pb₂I₆·2H₂O experimentally. These DFT calculations further reasonably demonstrate that (CDMA)Pb₂I₆·H₂O is not formed.

Fig. R2: Comparison of GIWAXS images of the DJ perovskite films.

According to the reviewer's comments, we have further redefined the (hkl) and investigated the possibility of a flat-on orientation of the slabs covering the surface leading to hydrophobicity, as shown in Figure 3c and 3d (Figures R2). On this basis, we further fabricated the $n = 1$ perovskite films, in which the inorganic slabs exhibit flat-on orientation features on the substrate, as depicted in Figures R3. XRD analysis shows that both films have the same flat-on orientation features. When these films were exposed to 85RH% conditions, PDMA-based films completely degrade in a few hours, while CDMA remains unchanged for more than 240 days. We further performed the contact angles of the water droplets on perovskite films to compare their hydrophobicity. The CDMA-based films show a larger water contact angle (47.13°) than the PDMA-based film (30.44°), indicating a relatively high water-resistant performance. The two have the same flat-on orientation features, but show large differences. It further confirms that it originates from the intrinsic stability, which is in consistent with the DFT calculations.

Fig. R3: a, Schematic diagram of perovskite on the substrate, green dot dash line represents the orientation of (100)/(200). b, Contact angles of the water droplets on perovskite films. c, d, XRD patterns of films exposed in a constant temperature humidity chamber with relative humidity (RH) of about 85% under dark at room temperature.

2. The definition of (hkl) does not comply with the crystal parameters of orthorhombic, as indicated in

Supplementary Table 6. In Figure 3c and 3d, only (111) and (311) match to the calculation of orthorhombic ($1/d^2=(h^2/a^2)+(k^2/b^2)+(l^2/c^2)$), but other do not. For example, q peaks of (00-2), (-202) and (-15-2) should be 0.2595, 1.54 and 3.636 A⁻¹.

✓ Response:

We are grateful for the comment. According to the reviewer's suggestions, we have redefined these mismatched (hkl), as shown in Figure 3c and 3d (Figures R2). The (hkl) were further defined by the following orthorhombic calculation formula:

$$\frac{1}{d^2} = \frac{h^2}{a^2} + \frac{k^2}{b^2} + \frac{l^2}{c^2}$$

Where q peaks of (110), (020), (214), (216), (220), and (310) are calculated to be 1.00, 1.42, 1.67, 1.75, 2.01, and 2.25 A⁻¹, respectively. These data are match well with the GIWAXS images and crystal parameters (Fig. R2). Based on this, we also updated the XRD results (**Supplementary Fig. 17**).

It is worthy to point out that there is no (111) peak in the PDMA-based perovskite according to the simulative XRD data of the PDMA single crystal (Fig. R4, please see attached CIF file and CCDC 2078954 for detailed crystal data). Although there are peaks of (111) in the CDMA perovskites, their intensities are relatively low, and the position of the q peak calculated by (110) is closer to the position in GIWAXS, so the definition of (hkl) is further changed to (110) instead of (111).

Fig. R4: Simulative XRD patterns of PDMA and CDMA single crystals.

3. The authors are unable to define (hkl) correctly, leading to an inability to clarify the orientation relationship between the crystal and the substrate. In the section of Characterization of the perovskite films, the authors mentioned that “In contrast, CDMA-based film shows relatively strong and discrete Bragg spots and out-of-plane distribution.... It suggests a high degree of the preferential vertical orientation of the 2D perovskite inorganic frameworks, forming a favorable charge-transport channel in the solar cells.” Based on the orthorhombic lattice (Supplementary Table 6), higher intensity of (111) does not indicate the preferential vertical orientation.

✓ Response:

Thank you for your precious comments. We have redefined these mismatched (hkl) and revised the corresponding discussions at Page 7 Line 3 in the revised manuscript. Details are as follows:

The grazing incidence wide-angle X-ray scattering (GIWAXS) were further studied as shown in Fig. 3c, Fig. 3d, and Supplementary Fig. 18. Compared with PDMA-based perovskite, CDMA-based film depicts relatively strong and discrete Bragg spots, which indicates the CDMA film has better crystallizations benefitting charge transport. The specific orientations assigned as (110), (020), (214), (216), (220), and (310) for both PDMA and CDMA-based DJ perovskites. Azimuthal scans of the (110) and (220) diffractions were further extracted from the GIWAXS measurements (Supplementary Fig. 18).

4. Figure S18 shows that CDMA has higher intensity of (111) diffraction than that of PDMA at the out-of-plane direction. However, Figure R1 presents a result opposite to those in Figure S18.

✓ Response:

We appreciate the reviewer's careful review. We double-checked the data and replaced the Supplementary Fig. 18 with the correct Fig. R5.

Fig. R5: Azimuthal scans of the (110) and (220) diffraction from GIWAXS measurements.

5. It should added FTIR date to demonstrate the hydrogen bond

✓ Response:

Thank you for your precious comments. We have added the FTIR data of $n=1$ CDMA- and PDMA-based DJ perovskites as shown Fig. R6 (Supplementary Fig. 37). In contrast to organic cations, both the N-H stretching and N-H bending peak of CDMA-based perovskite films show a distinct blue shift to relatively short wavelength, suggesting the relatively stronger interaction of CDMA cations and the inorganic framework by the hydrogen bond (ACS Appl. Mater. Interfaces 2022, 14, 51, 56900–56909; J. Phys. Chem. Lett. 2015, 6, 15, 2913–2918).

Fig. R6: FTIR spectra of the PDMA/CDMA cations and the corresponding $n=1$ CDMA- and PDMA-

based DJ perovskites.

6. In text, “The ultraviolet-visible (UV–Vis) absorption... DJ perovskite films (Supplementary Fig. 17b)”. The Fig. 17b should be corrected as Fig. 17c.

✓ Response:

We really appreciate the reviewer’s careful review. The text of Fig. 17b has been corrected as Fig. 17c.

7. Some X-ray data use q but some use 2θ , it is recommended to use q uniformly.

✓ Response:

Thank you for your suggestion. We have checked many references, where XRD diffraction commonly uses 2θ and GIWAXS generally uses q . If all q is replaced, it may cause trouble for the readers to compare the XRD diffraction data. Therefore, we want to be consistent with how the data is plotted in most of the references.

Reviewer #2 (Remarks to the Author):

The revised manuscript by Zhang and coworkers addresses the weak points that have been raised in the review process. The authors have replied convincingly to each of the criticisms and the manuscript, in my opinion, is significantly improved.

I recommend proceeding to publication.

✓ Response:

We sincerely thank you for your time and effort to improve our manuscript.

REVIEWERS' COMMENTS

Reviewer #1 (Remarks to the Author):

This manuscript has been completely revised and deserves to be published in nature communications.